# OpenReview forum: "Examining Alignment of Large Language Models through Representative Heuristics: the case of political stereotypes"
_ICLR.cc/2025/Conference — ICLR 2025 Poster_

### Official Review · Reviewer_pzsx · 2024-10-31

**Soundness:** 3
**Presentation:** 3
**Contribution:** 2
**Rating:** 6
**Confidence:** 3

**Summary:**

This paper explores the alignment of large language models (LLMs) with human intentions, focusing specifically on their susceptibility to political stereotypes. It investigates how LLMs deviate from empirical political positions, often exaggerating these positions compared to human respondents, which suggests vulnerability to representativeness heuristics. Experiments demonstrate that prompt-based mitigation strategies can reduce these tendencies, providing insights into better aligning LLMs with human values and reducing biased behavior.

**Strengths:**

1. The paper brings an underexplored perspective to understand and mitigate bias in LLMs by introducing representativeness heuristics from cognitive science in the context of political stereotypes.

2. It proposes a systematic quantification of the conditions under which LLMs deviate from empirical political positions, assessing the extent of bias and misalignment.

3. The mitigating strategies via prompt provide a simple yet practical solution to reduce stereotypes.

**Weaknesses:**

1. Presentation of the paper needs improvement. Some figures and tables are too small to read (i.e. Figures 3 and 4, Tables 1, 3, 7, and 8, etc.). The figure size is not consistent. The color denoted different methods in Figure 2 are hard to distinguish. There are some repeated definitions or sentences, such as the re-definition of kappa in the paragraph of **Prompt Style Mitigation Analysis**.

2. Lack of analysis of prompt style mitigating strategies’ results, such as which strategies make LLMs more aligned to human preferences, why baseline LLMs perform better in some tasks, etc.

3. The **potential effectiveness of political representative heuristics on downstream tasks** is unclear. The connection between stereotypes that this paper identifies and quantifies to fake news should be more clearly explained. The behavior of LLMs in fake news detection could be affected by the pre-training corpus.

**Questions:**

None

---

> ### Author Response · Authors · 2024-11-19
>
> Dear Reviewer pzsx,
>
> Thank you for your thoughtful and constructive feedback. We greatly appreciate your time and effort in reviewing our paper. Below, we provide detailed responses to your comments and outline the revisions we have made in response.
>
> **Presentation Improvement**: We have made significant improvements to the presentation of our tables and figures to ensure they are clear, concise, and easy to interpret. Additionally, we have streamlined the "Prompt Style Mitigation Analysis" section by removing repetitive definitions and sentences.
>
> **Prompt Style Mitigation Analysis**: In the revised results section, we have illustrated the effects of the various prompt style mitigation strategies. As expected, the highest κ values (indicating stronger stereotyping) were observed in the baseline case, where no mitigation strategies were applied. This suggests that, in the absence of intervention, models tend to exhibit higher levels of stereotyping. The effectiveness of the mitigation strategies varied across tasks and models.
>
> **Relation to Downstream Tasks (Misinformation Detection**): We acknowledge that the relation to downstream tasks may not be clear. We focused on how party affiliation information—which encodes representative characteristics of the political parties—might act as a proxy influencing model performance on downstream tasks like misinformation detection. In the controlled experiment, we sought to investigate whether the inclusion of party affiliation affects the model’s ability to detect fake news. We note that this experiment does not establish a causal relationship between representative heuristics and the model’s performance in downstream tasks. Rather, it serves as an exploratory analysis to examine whether party affiliation information impacts the model’s performance in misinformation detection within a controlled experimental setting. We hope this clarifies the purpose and limitations of this experiment.
>
> Once again, we sincerely appreciate your careful reading of our paper and the valuable feedback you have provided. We believe that the revisions we have made have strengthened the paper, and we hope that our clarifications and improvements address your concerns. Should you have any further questions or suggestions, please do not hesitate to reach out.
>
> Thank you again for your time and consideration.

---

> > ### Comment · Reviewer_pzsx · 2024-11-24
> >
> > I thank the authors for their response. I have increased my score for "Presentation" as the authors did significant work to improve it in the rebuttal submission. This also led me to increase the overall assessment score.
> >
> > I still have some reservations about the impact on the downstream tasks, as political biases in LLMs are documented in previous studies. If other reviewers suggest acceptance, the paper should be accepted.

---

### Official Review · Reviewer_3joA · 2024-11-03

**Soundness:** 3
**Presentation:** 3
**Contribution:** 3
**Rating:** 6
**Confidence:** 4

**Summary:**

This paper examines the alignment of LLMs through representative heuristics using political stereotypes as a reference context. The authors unveil that although LLMs can mimic certain political parties' positions on specific topics, they do so in a more exaggerated manner compared to humans. Finally, this work proposes some prompt-based mitigation strategies aimed at limiting such exaggerations.

**Strengths:**

- The findings of this work are valuable, as the unveiling of exaggerated positions compared to humans (despite being limited on the political context) is key to better comprehending how we should interact with these systems, and whether interventions are needed to align them more with human values and perspectives.
- The manuscript is well written, the methodology is properly formalized, non-ambiguous, and easy to follow. All methodological aspects are well supported by reference literature.
- The choice for diverse LLM families is valuable as sheds light on the different "behaviors" they might exhibit based on varying training data and alignment approaches.
- The proposed intervention techniques turn out to be reasonably effective in mitigating the exaggerated intrinsic behaviors.
- The Appendix of the manuscript complements the main content with additional relevant information for the proper understanding of the work.

**Weaknesses:**

- Focusing just on a single context (i.e., political) and scenario (the US one) is the weakest point to me, as it limits the generalizability of the unveiled patterns.
- Despite being valuable, the results would require more emphasis on the conditions underlying certain behaviors (as stated throughout the manuscript), as it will further help this work unveil the roots of the unveiled exaggerations.
- The results presentation contrasts with the methodology, as it has room for improvement in both the figures/tables presentation (some of them are hard to read) and discussion.

**Questions:**

- Adding more up-to-date models would be useful to also grasp potential "developments" into the unveiled positions; similarly, considering some open models might improve matching certain behaviors with specific approaches (thanks to potentially greater transparency in training data and alignment techniques).
- As the authors mentioned refusals, I wonder how they handled them and on what occasions they occurred. Shedding light on the latter point would further unveil the roots of certain exaggerated positions.
- Related to the previous point, did the models experience hallucinations? If yes, how were they handled?
- As a minor remark, Section 11 might contain some typos on the followed Ethics Policy.

---

> ### Author Response · Authors · 2024-11-19
>
> Dear Reviewer 3joA,
>
> Thank you very much for your thoughtful and constructive feedback. We appreciate the time and effort you have put into reviewing our work, and we have carefully considered each of your points. Below, we provide detailed responses to your comments.
>
> **Generalizability**: We fully acknowledge the importance of generalizability in this research. While the primary focus of this paper is on political stereotypes, the methodology employed in our analysis can indeed be extended to other domains. For example, datasets such as  GlobalOpinionsQA [1], OpinionQA Dataset [2] offer empirical data on global representations, and the methods outlined in our paper could easily be adapted to analyze these datasets. By doing so, researchers could investigate the representative heuristic behaviors of large language models (LLMs) across different domains, which would provide further insight into their generalizability.
>
> [1] Durmus, E., Nyugen, K., Liao, T. I., Schiefer, N., Askell, A., Bakhtin, A., ... & Ganguli, D. (2023). Towards measuring the representation of subjective global opinions in language models. arXiv preprint arXiv:2306.16388.
>
> [2] Santurkar, S., Durmus, E., Ladhak, F., Lee, C., Liang, P., & Hashimoto, T. (2023, July). Whose opinions do language models reflect? In International Conference on Machine Learning (pp. 29971-30004). PMLR.
>
> **Roots of exaggerations**: While the primary aim of our paper was not to explore the underlying causes of exaggerations in LLMs' responses, we recognize that this is an important issue. In Appendix H, titled "Aligning Methods and Representative Heuristics," we provide an initial analysis comparing the base model with the RLHF-trained model. Our observations suggest that RLHF, which is typically considered a process to mitigate harmful biases and enhance helpfulness, might unintentionally exacerbate representative heuristic-based stereotypes. Specifically, it appears that RLHF could push the model toward exaggerating beliefs about certain political groups. However, we acknowledge that the RLHF phase is influenced by multiple confounding factors, such as the training dataset and the algorithms used. Therefore, we believe that further research on the interplay between RLHF and heuristic based exaggeration would be valuable. However, we have left this exploration outside the scope of the current paper, as we aimed to focus primarily on the methodological aspects.
>
> **Figures and Table Presentation**: We have carefully revised and improved the presentation of our figures and tables to enhance clarity and readability. We believe the changes make the results more accessible and will provide readers with a clearer understanding of our findings. Please refer to the updated submission for the revised tables and figures.
>
> **Including More Open Models**: We completely agree with your suggestion to include more open models. In response, we have added recent open models, specifically Llama 3-8b and Qwen 2.5-72b, to our analysis. We hope this addition further enriches the scope and relevance of our findings.
>
> **Handling Refusals**: Refusal responses occurred in a few specific instances, particularly when querying Gemini about sensitive topics such as "Government Aid for Blacks" within the ANES dataset. We suspect these refusals are a result of automatic regulations within the Gemini model, which may reject queries containing sensitive terms, such as those related to race or ethnicity. To ensure the integrity of our analysis, we excluded any instances where refusal responses were generated.
>
> **Hallucinations**: As this task is focused on subjective opinions rather than objective factual questions, we did not observe hallucinations in the generated outputs. However, to assess the quality and relevance of the responses, we conducted a human evaluation, as detailed in Appendix E: Human Evaluation Analysis. This evaluation allowed us to confirm that the generated responses were relevant to the queries, despite the subjective nature of the task.
>
> **Minor Revision on Ethics Policy**: Thank you for pointing out the need for revision in our ethics policy section. We have updated this section.
>
> Once again, we deeply appreciate the time and effort you have invested in reviewing our paper. We hope that the revisions we have made address your concerns satisfactorily. If you have any further questions or suggestions, please do not hesitate to reach out.

---

> > ### Comment · Reviewer_3joA · 2024-11-25
> >
> > I thank the authors for their detailed response. I have increased the presentation score as the authors have done substantial work to improve the overall presentation of the manuscript, which is now qualitatively adequate.
> >
> > I will maintain my overall score due to remaining concerns about the generalizability of the findings to other contexts/scenarios.
> > Nonetheless, I will support acceptance if other reviewers agree on this.

---

### Official Review · Reviewer_nJzF · 2024-11-04

**Soundness:** 4
**Presentation:** 3
**Contribution:** 3
**Rating:** 8
**Confidence:** 3

**Summary:**

This paper focuses on the challenges and limitations of using LLMs to simulate human behaviour. In particular, it discusses how LLMs measure stereotypical behaviour w.r.t. groups of individuals self-identified as either Democrats or Republicans. The authors use GPT-3.5, GPT-4, Gemini Pro, and Llama2 models to estimate to what extent the beliefs generated by LLMs are representative of aggregated empirical opinions specified by individuals belonging to either party (the authors use two existing datasets, ANES and MFQ, for their analysis). Results show that for ANES, LLMs tend to inflate responses for Republicans, and deflate responses for Democrats. The same is true for Democrats on MFQ (the results for Republicans are inconsistent). Overall, the results show that beliefs are consistently exaggerated by LLMs as compared to the empirical means derived from human surveys.

**Strengths:**

* The paper discusses an interesting topic by analyzing to what extent LLM responses are representative of human responses in the context of political opinions. The provided results are useful to inform future work aiming to better understand how LLMs can be used in that context.
* The paper’s analysis is overall extensive and thorough, even though I have recommendations on improving the paper's structure (see weaknesses).
* I appreciate the Limitations specified in Section 10 of the paper.

**Weaknesses:**

* The paper uses excessive formalism to introduce the proposed method and several crucial details are moved into the Appendix. To improve readability and presentation of the obtained findings, I’d recommend to move parts of Section 3 into the Appendix instead, and add more details on the empirical setup to the main manuscript.
* The presentation could be improved. Citations should be surrounded with parentheses if used passively as this improves readability. Some citations in Section 5.2 are incorrectly ordered. The results in Figure 2 could be presented more clearly, for example by disentangling the plots between Democrats and Republicans. I find some of the Tables (e.g., Table 1 and 3) too full and overwhelming.

**Questions:**

On the prompt sensitivity check in Appendix F, do you have an understanding of how this changes when adjusting the temperature values? Or, more generally, how much variation in the obtained results would you expect as the temperature values provided in Appendix D change?

---

> ### Author Response · Authors · 2024-11-19
>
> Dear Reviewer nJzF,
>
> We sincerely thank you for your thoughtful and valuable feedback.
>
> **Readability and Presentation**: We have made significant efforts to simplify the presentation of our results, while still ensuring that the key findings are clearly conveyed. These changes have been applied to all tables and figures in the main text.
>
> **Citation Format**: We have updated the citation format to use parentheses.
>
> **Figure 2**: In response to your recommendation, we have revised Figure 2 by separating the data for Democrats and Republicans for greater clarity. Additionally, we have simplified Table 1 and Table 3 to improve readability.
>
> **Temperature Sensitivity Analysis**: We have conducted a temperature sensitivity analysis and included the results in Appendix E ("Sensitivity Check of Prompts"). Specifically, to assess temperature sensitivity, we ran GPT-4 on the *Anes* task 10 times for each temperature setting. For each topic, we computed the Coefficient of Variation (CV) and averaged the results. The `Diff_D` represents the difference between the Believed Mean of Democrats and the Empirical Mean, while `Diff_R` reflects the difference between the Believed Mean of Republicans and the Empirical Mean. The results show that the CV increases with higher temperature settings, indicating greater variability in the responses. However, when averaged, the deviations from the empirical mean (`Diff_D` and `Diff_R`) remain relatively consistent, with values around -1.4 and 0.46, respectively.
>
> | Temperature | 0 | 1 | 1.5 | 2 |
> |--------------------------|-------|-------|-------|-------|
> | **Coefficient of Variation** | 0.00 | 0.03 | 0.06 | 0.11 |
> | **Diff_D** | -1.51 | -1.46 | -1.40 | -1.40 |
>  | **Diff_R** | 0.48 | 0.46 | 0.49 | 0.47 |
>
> Once again, we truly appreciate the time and effort you dedicated to reviewing our paper. We hope we have adequately addressed your concerns, but please feel free to reach out if there are any further issues or clarifications needed.

---

> > ### Comment · Reviewer_nJzF · 2024-11-25
> >
> > Many thanks to the authors for the detailed reply. I'll keep my score indicating acceptance.

---

### Author Response · Authors · 2024-11-19

We'd like to sincerely thank the reviewers for their insightful comments and valuable feedback. We appreciate the positive remarks recognizing our work as both an interesting topic with thorough analysis (Reviewer nJzF), a valuable contribution (Reviewer 3joA), and one offering underexplored perspectives (Reviewer pzsx). We have made every effort to address the points raised.

**Enhancing Readability**:
In response to common feedback, we have revised the Tables and Figures to improve clarity. Specifically, we have aggregated results by dataset to simplify the presentation. Detailed results have been moved to the appendix, allowing readers to access disaggregated results while enhancing the readability of the main text. Please refer to the revised submission for these updates.

**Adding More Open Models**:
We have also added recent open-source models, such as Llama 3-8b and Qwen 2.5-72b, to broaden the scope of our analysis. We believe that the transparency of these models—offering open training data and code—will support further research on this topic.

Thank you again for your valuable feedback. We hope these revisions address your concerns and improve the clarity of the paper.

---

### Meta-Review · Area_Chair_zQ1K · 2024-12-08

**Metareview:**

This paper investigates the extent to which large language models (LLMs) align with human responses in the context of political stereotypes. Using datasets like ANES and MFQ, the authors analyze how LLMs simulate political opinions and highlight their tendency to exaggerate group-specific beliefs compared to empirical human data. The study evaluates multiple LLMs (e.g., GPT-3.5, GPT-4, Gemini Pro, Llama2) and introduces prompt-based mitigation strategies to reduce these exaggerations. The results contribute to understanding and improving LLM alignment with human values.

This paper offers a thorough and well-structured analysis of LLM alignment with human values, focusing on political stereotypes. While the scope is limited to a specific context, the methodology, findings, and proposed mitigation strategies are highly valuable for future research in bias mitigation and model alignment. Addressing the presentation and analysis depth in a revision would make the paper even stronger. I recommend acceptance with minor revisions.

**Additional Comments On Reviewer Discussion:**

The discussion during the rebuttal period mainly focuses on the additional experiments such as more LLMs and parameter analyses which were addressed by the authors, and on the clarity of the paper writing. Authors are encouraged to integrate the new experiments and improve paper writing in future versions.

---

### Decision · Program_Chairs · 2025-01-22

Accept (Poster)